# Role of the Transcription Factor MAFA in the Maintenance of Pancreatic β-Cells

**DOI:** 10.3390/ijms23094478

**Published:** 2022-04-19

**Authors:** Wataru Nishimura, Hiroaki Iwasa, Munkhtuya Tumurkhuu

**Affiliations:** 1Department of Molecular Biology, School of Medicine, International University of Health and Welfare, Narita 286-8686, Chiba, Japan; iwasah@iuhw.ac.jp; 2Division of Anatomy, Bio-Imaging and Neuro-Cell Science, Jichi Medical University, Shimotsuke 329-0498, Tochigi, Japan; 3Department of Genetics and Molecular Biology, School of Bio-Medicine, Mongolian National University of Medical Sciences, Ulaanbaatar 14210, Mongolia; munkhtua.t@mnums.edu.mn

**Keywords:** MAFA, MAFB, insulin, β-cells, diabetes mellitus, plasticity, dedifferentiation, transcription factor

## Abstract

Pancreatic β-cells are specialized to properly regulate blood glucose. Maintenance of the mature β-cell phenotype is critical for glucose metabolism, and β-cell failure results in diabetes mellitus. Recent studies provide strong evidence that the mature phenotype of β-cells is maintained by several transcription factors. These factors are also required for β-cell differentiation from endocrine precursors or maturation from immature β-cells during pancreatic development. Because the reduction or loss of these factors leads to β-cell failure and diabetes, inducing the upregulation or inhibiting downregulation of these transcription factors would be beneficial for studies in both diabetes and stem cell biology. Here, we discuss one such factor, i.e., the transcription factor MAFA. MAFA is a basic leucine zipper family transcription factor that can activate the expression of insulin in β-cells with PDX1 and NEUROD1. MAFA is indeed indispensable for the maintenance of not only insulin expression but also function of adult β-cells. With loss of MAFA in type 2 diabetes, β-cells cannot maintain their mature phenotype and are dedifferentiated. In this review, we first briefly summarize the functional roles of MAFA in β-cells and then mainly focus on the molecular mechanism of cell fate conversion regulated by MAFA.

## 1. Introduction

The total population of patients with diabetes worldwide is predicted to reach 537 million in 2021 and 783 million in 2045 [1]. Pancreatic β-cells secrete insulin to regulate blood glucose. The dysfunction and/or reduced mass of pancreatic β-cells results in impaired glucose-stimulated insulin secretion (GSIS), leading to diabetes. Therefore, it is important to clarify the molecular mechanism of β-cell failure for elucidation of the pathophysiology of diabetes.

Recent studies challenge the concept of cell fate determination, suggesting that cell differentiation is a dynamic state [2,3]. Many studies in various tissues have demonstrated that the manipulation of a few master transcription factors can specifically allow somatic cell fate conversion into a particular cell type in vivo [4,5,6,7]. The plasticity of somatic cells in a pathological state, with loss of these transcription factors, has been intensively investigated in pancreatic endocrine cells. Recent research shows that the molecular mechanism of β-cell failure in type 2 diabetes involves identity loss or dedifferentiation. Lineage tracing studies have demonstrated that several transcription factors critical for β-cell differentiation or maturation [8,9,10,11], including v-Maf musculoaponeurotic fibrosarcoma oncogene family transcription factor A (MAFA) [12], control the maintenance of the mature phenotype of β-cells. The current review primarily focuses on the regulation of pancreatic β-cell maintenance by MAFA after a brief summary of the role of MAFA in the development and function of pancreatic β-cells with the purpose of enabling the development of novel strategies to treat diabetes by preserving or improving β-cell function. An overview of β-cell dedifferentiation has been explained in previously published reviews [13,14,15,16,17].

## 2. Targets of the Transcription Factor MAFA

The insulin 2 promoter in rat pancreatic β-cells is regulated by three major elements, A3, C1-A2 and E1 [18,19,20]. Mutation in any of these elements resulted in reduction in the promoter activity of rat *Ins2*, suggesting the importance of transcription factors that can bind to these elements [21]. MAFA was identified as a binding factor and an activator of the C1-A2 element [22]. An electrophoretic mobility-shift assay using a probe from −139 to −101 bp of the rat *Ins2* promoter successfully detected a 47 kDa protein from the HIT T-15 hamster insulinoma cell line, which was identified as MAFA by mass spectrometry. This finding was later confirmed by three independent studies [23,24,25]. Meanwhile, PDX1, a homeodomain transcription factor, and NEUROD1, a transcription factor that belongs to the basic helix-loop-helix family, can bind A3 and E1, respectively [18,19,21]. These three factors can interact with each other and cooperatively activate the insulin gene in β-cells [26,27].

The glucose-dependent expression of MAFA in β-cells [20] prompted researchers to investigate the function of MAFA in β-cells. Accumulating evidence has revealed that MAFA is critical for the expression of not only *Ins2* but also *Ins1*, *Slc2a2*, *Slc30a8*, *Pcsk1*, *Pdx1*, *Sytl4*, *Maob*, *Vdr*, *Prlr*, *Ccnd2*, *Ucn3* and *ChrnB4* in β-cells [28,29,30,31,32,33,34]. Most of these molecules are involved in GSIS and play a functional role in mature β-cells. The results of another study have also revealed that MLL3/4 function as transcriptional coactivators of MAFA and play a role in inducing the expression of *Ins2*, *Slc2a2*, *G6pc2*, *Slc30a8* and *Ccnd2* in β-cells [35]. Recent studies revealed that the expression of exocytosis-related genes *Stx1a* and *Stxbp1*, subunits of voltage-gated Ca^2+^ channels *CaVγ4*, and *Ppp1r1a* that is involved in GLP1R-mediated amplification of GSIS, is also regulated by MAFA, further demonstrating the importance of MAFA in insulin secretion [36,37,38]. Analysis of islet-specific enhancers by ChIP-seq of mouse islets revealed 3638 MAFA-enriched loci [34].

So far, the analysis of global [12,28,29] and pancreas-specific *Mafa* knockout mice [30] has been reported. These mice have similar phenotype, showing impaired mass and function of β-cells by 3 to 4 weeks of age, reduced proliferation with no accelerated apoptosis of β-cells, downregulation in the expression of *Ins1*, *Ins2*, *Slc2a2*, *Slc30a8* and *Pdx1* in their islets, and glucose intolerance. Transcriptome analyses of islets isolated from *Mafa* knockout mice have further elucidated candidate molecules that are regulated by MAFA. Genes downregulated in both knockout mice include *Trpm5*, *Sytl4*, *Slc14a2*, *BC039632*, *Gad1*, *Maob*, *Ttc28*, *Lifr*, *Rhobtb1*, *Slc2a2*, *Car10*, *Atp7a*, *Papss2*, *Scel*, *Prlr*, *F13a1*, *Nup93*, *Slc30a8*, *Vdr*, *Ccnd2*, *Cryl1*, *Bag2*, *Ucn3*, *Coro2b* [29,30]. In addition, it has been recently reported that MAFA plays a role in the inhibition of cytokine production from β-cells, which is involved in islet inflammation [39].

## 3. The Role of Maf Factors in the Developing Pancreas

PDX1 is expressed in early pancreatic buds at E8.5. *Pdx1* knockout embryos are apancreatic [40]. The expression of NEUROD1 is restricted to the endocrine cells of the pancreas. A striking reduction in the number of endocrine cells is observed in the pancreases of *Neurod1* knockout embryos [41]. These data underscore the importance of insulin gene transcription factors in pancreatic development. The expression of MAFA occurs during pancreatic development starting at E12.5 to E13.5 and can be observed exclusively in insulin-expressing (insulin^+^) cells [42,43]. Meanwhile, another large Maf factor MAFB is expressed in glucagon^+^ cells as early as at E10.5 and also in insulin^+^ cells prior to MAFA during embryonic development of murine pancreas [42,43]. Interestingly, MAFB shares a DNA-binding region with MAFA and thus can bind the C1-A2 element of insulin promoter in vivo, activate it in vitro, and can also bind R3 region of *Mafa* promoter during β-cell development [44,45]. MAFB expression persisted in glucagon^+^ and insulin^+^ cells but not in either somatostatin^+^ or pancreatic polypeptide^+^ cells during development, which is gradually restricted to glucagon^+^ cells after birth, and it is selectively expressed in the glucagon-producing α-cells of the adult pancreatic islets [42,44,46]. β-cell-specific and α-cell-specific expression of MAFA and MAFB in adult mice pancreas, respectively, have been confirmed not only by immunohistochemistry but also by promoter activities of *Mafa* and *Mafb* that drive the expression of two fluorescent proteins independently in mice [47]. Numerous studies have demonstrated that immature β-cells express MAFB, while mature β-cells express MAFA in the embryonic or neonatal pancreas [42,44,48,49]. The changes in the expression of Maf factors during development of the pancreas indicate that the terminal differentiation process toward mature β-cells occurs even after the expression of insulin. These observations are further supported by the results of stem cell studies showing that the ability to secrete insulin from ES-derived insulin^+^ cells accompanied the expression of MAFA [50,51,52]. In adult islets, the expression of MAFA is not homogenous [42], which has been validated by recent single-cell analyses [53] and reveals the heterogeneity of islet cells, indicating that transcriptionally mature and immature β-cells coexist within the adult islet together [54]. Other than MAFA, UCN3 is recognized as a marker for mature β-cells, although the genetic deletion of *Ucn3* does not cause a loss of β-cell maturity or an increase in β-cell dedifferentiation [55], suggesting the importance of MAFA as a marker of mature β-cells.

*Mafb*-deficient pancreas have a reduced number of insulin+ and glucagon+ cells with reduced expression of PDX1 and MAFA, without affecting endocrine progenitor cells expressing NEUROG3, NKX2-2, NKX6-1 and PAX6 [44,49]. *Pax6*-deficient pancreas have a similar phenotype, but the expression of MAFB is downregulated [49]. In contrast with these mutant embryos that have a reduced number of insulin^+^ cells, the embryonic development is normal in the *Mafa^−/−^* pancreas [12,28,56]. Meanwhile, the overexpression of MAFA in *Pdx1*-expressing cells in the early pancreatic bud does not convert these cells into insulin^+^ cells but inhibits differentiation and proliferation, suggesting that the sequential activation of the expression of transcription factors is critical for endocrine differentiation in the embryonic pancreas [57].

In addition to these data, it is intriguing that the ectopic expression of MAFA, PDX1 and NEUROD1 (or NGN3) converts adult liver or pancreatic acinar cells to β-cells [4,58,59]. These three transcription factors may be master genes of pancreatic β-cells, which can induce the expression of genes necessary for cell fate conversion.

## 4. The Role of MAFA in the Maintenance of the Mature β-Cell Phenotype

Numerous studies have demonstrated that the expression of MAFA is impaired in β-cells of rodents and humans with diabetes [60,61,62,63]. This reduction in the expression of MAFA in compromised β-cells occurs prior to the downregulation of other transcription factors that are expressed in β-cells, such as PDX1 and NKX6-1 [60]. The loss of MAFA results in the reduced expression of molecules that are critical for the function of β-cells, as discussed above [12,28,29,30,56].

In addition, accumulating evidence suggests that MAFA is not only critical for insulin biosynthesis and GSIS but also indispensable for maintenance of the mature phenotype of β-cells [12,64,65]. Although *Mafa*-deficient mice have a comparable number of β-cells throughout embryonic development and at birth, they become intolerant to glucose with reduced or no expression of insulin in the islets, although they do not show overt diabetes [12,28,56]. The β-cell to α-cell ratio in the islets of the pancreas decreases during the neonatal period. Importantly, genetic lineage tracing analysis revealed that *Mafa*^−/−^ β-cells retain an endocrine cell phenotype with the expression of SYP and CHGA but have reduced or lost the expression of insulin, although a few expressed glucagon. These insulin-negative “empty” endocrine cells in *Mafa*^−/−^ islets have decreased expression of molecules critical for β-cell function, such as *Ins1*, *Glut2*, *Slc30a8*, *Pcsk1*, *Vdr* and *Ucn3*, as well as increased expression of molecules conventionally repressed in β-cells, such as *Gcg*, *Mafb*, *Pax4*, *Neurog3*, *Sox9*, *Sox2*, *Nanog* and *Mct1*, some of which are recently identified as β-cell disallowed genes [66,67]. These phenomena are now recognized as the dedifferentiation of β-cells [9,12], which has been described earlier [3,68] and validated by lineage tracing [69]. Not only in *Mafa*-deficient mice but also diabetic mice with reduced expression of MAFA have a deeper loss of β-cell identity with the changes in gene expression above [12]. These results suggest that MAFA is critical for the formation and maintenance of the mature β-cell phenotype and that dedifferentiation with a loss of MAFA could be the common mechanism of β-cell dysfunction in type 2 diabetes in both mice and humans.

In the rodent study, this ‘loss of β-cell identity’ is characterized by (1) the decreased or completely absent biosynthesis of insulin in β-cells demonstrated by genetic lineage tracing studies, (2) the existence of “empty” endocrine cells in islets shown by electron microscopy and (3) the impaired expression of genes critical for β-cell function with increased expression of molecules that are normally repressed in β-cells, including the upregulation of transcription factors that are transiently expressed in endocrine precursors such as the ‘immature β-cell marker’ MAFB. A certain fraction of β-cells with loss of identity are transdifferentiated to glucagon^+^ cells. These observations can also be seen in *Foxo1*, *Pdx1*, *Pax6* and *Nkx2-2* knockout mice (Table 1) [8,9,10,11,12,70], most of which are also important for β-cell specification during pancreatic development [67,68,69,70,71]. In dedifferentiated β-cells, the increased expression of genes such as the transcription factors *Mafb* and *Arx*, which are critical for α-cell specification, may be induced by epigenetic modifications such as promoter demethylation [72,73]. Interestingly, the deletion of *Mafb* in diet-induced obese *Mafa*-deficient mice shows impaired islet formation, a decreased number of β-cells and diabetes, which is more advanced than those in diet-induced obese *Mafa*-deficient mice, suggesting that MAFB may have a role in the maintenance of adult β-cells with a reduced expression of MAFA [74], although another study demonstrated that MAFB alone was unable to rescue the β-cell defects in mice lacking *Mafa* [75]. Taken together, MAFA is critical for the fate of β-cells in adult pancreas.

Recent studies have also shown that there is a redifferentiation of β-cells via intensive insulin therapy in a diabetes mice [10]. This phenomenon may also take place in humans and contribute to the recovery of insulin secretion that has been observed in diabetic patients who have undergone intensive insulin therapy [76]. These results raise the possibility that factors that can upregulate MAFA or inhibit MAFA downregulation may induce the redifferentiation of β-cells in individuals with diabetes. Indeed, the expression of MAFA induced by a Cre-loxP-Rosa system in β-cells of diabetes model mice increased plasma insulin, ameliorated elevated blood glucose and HbA1c, and preserved β-cell function [76].

## 5. Factors Regulating MAFA Expression

MAFA is critical for the maintenance of β-cells, and its downregulation results in β-cell dedifferentiation in type 2 diabetes, as mentioned above. Several screening studies have identified small molecules that can preserve or improve β-cell function by upregulating PDX1 or UCN3 [77,78]. Inducing MAFA expression in individuals with diabetes could maintain the mature and functional β-cells. The identification of factors or small molecules that can modify MAFA expression may enable preserving or improving β-cell function, even if only temporarily. Thus, the upstream mechanisms by which MAFA expression or function is altered in β-cells have been investigated. So far, several factors have been identified that directly regulate MAFA expression.

### 5.1. Signal Transduction

The most intensively analyzed factor that compromises MAFA expression is oxidative stress. β-cells have relatively low levels of antioxidant enzymes and thus are sensitive to reactive oxygen species induced by hyperglycemia. Several studies have demonstrated that during chronic hyperglycemia, oxidative stress is induced in β-cells and compromises the expression of MAFA, which leads to reduced biosynthesis and secretion of insulin [58]. This impairment in insulin secretion from β-cells causes further hyperglycemia [79]. In these β-cells with so-called glucose toxicity, the reduction in MAFA expression caused by oxidative stress is mediated via an increased expression of c-JUN [80]. Oxidative stress can also promote the translocation of MAFA from the nucleus to the cytoplasm [60]. Recent study suggests that oxidative stress-induced β-cell failure may result from partial dedifferentiation with loss of *MAFA* and *PDX1* and concomitant upregulation in the expression of progenitor markers *SOX9* and *HES1* [81]. These results are further supported by studies analyzing transgenic mice with β-cell-specific expression of thioredoxin or glutathione peroxidase, which are both redox proteins. These proteins can function as antioxidants to repair molecules that are oxidized by reactive oxidative intermediates, which can preserve the expression of MAFA and insulin, and improve β-cell function in diabetes mouse models [82,83]. These results raise the possibility that antioxidant drugs can improve β-cell function and hyperglycemia [84]. In addition to c-JUN, which is induced by oxidative stress, another important factor that can deteriorate β-cell function via the downregulation of MAFA during diabetes is TXNIP, a cellular redox regulator. TXNIP is upregulated in β-cells during diabetes and has been shown to induce *miR-204* by the inhibition of STAT3 phosphorylation, which in turn directly inhibits MAFA and insulin expression [85]. Another approach to ameliorate glucotoxicity is to use an SGLT2 inhibitor, which has also demonstrated protective effects on the expression of *Mafa* in β-cells in diabetic mice and has also been shown to maintain the expression of *Pdx1*, *Ins1*, *Slc2a2* and *Glp1r* [86,87]. In addition to oxidative stress, hypoxic stress in β-cells downregulates *Mafa* and other factors, such as *Pdx1*, *Foxa2*, *Neurod1*, *Ins1*, *Wfs1* and *Slc2a2* [88].

Another critical mechanism for the regulation of MAFA expression is TGF-β/BMP signaling, which downregulates MAFA through the Smad family and impairs β-cell maturity under mild metabolic stress conditions [89,90]. It has been shown that TGF-β signaling inhibitors can induce the activation of MAFA expression and the terminal differentiation of β-cells [78,91]. Conversely, mTOR signaling activates MAFA expression. Inhibition of the mTOR by Rapamycin or Tacrolimus impairs the expression of MAFA and insulin content [92]. Knockout mice of *Raptor*, an essential component of mTORC1, show reduced glucose responsiveness. Islets isolated from these mice have a reduced expression of *Mafa* and upregulation of neonatal markers and β-cell disallowed genes, resulting in a loss of functional maturity of β-cells [93]. FOXO1, a downstream factor of insulin signaling, protects β-cells against oxidative stress by forming a complex with PML and SIRT1 to activate the expression of *Mafa* [61]. Mice lacking functional receptors for both insulin and IGF-1 in β-cells have a reduced expression of phosphorylated AKT and MAFA [63].

The protein stability of MAFA is regulated by its phosphorylation, which is also involved in β-cell dysfunction in diabetes [62,94]. MAFA is a phosphorylated protein that is regulated by kinases such as GSK3 and MAPK14. The degradation of phosphorylated MAFA in β-cells with oxidative stress is regulated in a different manner from those without [95]. It has also been reported that JNK, which is induced by oxidative stress, and MAPK14 are involved in aldosterone-induced β-cell dysfunction; glucocorticoid receptors are stimulated by aldosterone and phosphorylate JNK and MAPK14, which in turn promote the degradation of MAFA [96]. Recently, it has been suggested that FERMT2, which belongs to the fermitin family, binds to and stabilizes MAFA, which activates insulin expression [97].

### 5.2. Noncoding RNA

Several noncoding RNAs are involved in the regulation of *Mafa* in β-cells. Other than *miR-204* induced by TXNIP, it has also been reported that *miR-149* impairs *Mafa* expression [98]. Another microRNA *miR-24* targets *Mafa* and impairs GSIS, while preserving β-cell mass by inhibiting apoptosis and inducing dedifferentiation [99]. The long noncoding RNA *Meg3*, *Gas5* and *lncRNA-p3134* are involved in β-cell function via the regulation of *Mafa*, although a detailed mechanism has not yet been shown [100,101,102,103].

### 5.3. DNA Binding Factors

The *Mafa* promoter is regulated by R3, which is located −8118 to −7750 bp from the transcription start site, where PDX1, FOXA2, NKX2-2, PAX4, PAX6, NKX6-1, NEUROD1, ISL1 and MAFB can bind [104]. These factors may be involved in the upregulation of *Mafa* during embryonic development of the pancreas. Among these factors, ISL1 has received attention as an activator of *Mafa* transcription. NRD1, an N-arginine dibasic convertase, can recruit ISL1 to R3 of the *Mafa* promoter and can induce its activation and *Mafa* expression [105]. SSBP3 and LDB1 have also been shown to function as coactivators of ISL1 for the direct regulation of the *Mafa* promoter [106,107]. Another promising factor for upregulating *Mafa* is PDX1, which can directly bind and strongly activate the *Mafa* promoter [104]. Indeed, a small molecule that can activate *Pdx1* expression can also upregulate *Mafa* [77]. In addition to these conventional factors that can bind and activate R3, a recent study demonstrated that CREB and its coactivator CRTC2 can stimulate *Mafa* expression by directly binding to the *Mafa* promoter at −1342 to −1346 bp [108], which contributes to insulin expression [109]. Islet RNA-seq combined with ChIP-seq analysis shows that transcription factor GLIS3 directly regulates the expression of MAFA and insulin [110]. Meanwhile, ONECUT1/HNF6 can bind to the FOXA2 binding site of the *Mafa* promoter and directly inhibit *Mafa* expression, and this inhibition may be implicated in *Mafa* expression in the embryonic pancreas, which is relatively late compared with the expression of other transcription factors during development [111]. MEN1 has also been shown to bind *Mafa* promoter to suppress its expression [112].

### 5.4. Hormones

Another remarkable factor that has recently been shown to upregulate MAFA is thyroid hormone. Triiodothyronine (T3) can enhance the expression of MAFA by directly activating thyroid hormone response elements of the *Mafa* promoter located at −1927 to −1946 bp and +647 to +659 bp and *Mafa* transcription, which has been implicated in the functional maturation of postnatal β-cells [113]. Additional studies have shown that T3, along with inhibitors of AXL and TGFBR1, can drive the terminal differentiation of stem cells into functional β-cells [114]. Pancreatic progenitors transplanted into mice with chronic hypothyroidism cannot mature when the expression of *Mafa* is reduced [115]. In addition, GLP-1 and its analogs can improve the function of β-cells, and they also accompany the upregulation of *Mafa* [116].

### 5.5. Other Factors

During the regeneration process of β-cells, REG3D, islet neogenesis-associated protein, is involved in the expression of *Mafa*, although the direct role and precise mechanism on *Mafa* expression is not clear [117]. It has also been reported that microenvironments, which include neurons [118], immune cells [119] and endothelial cells that surround the islets [120], are also involved in β-cell mass and function through *Mafa*. Pericytes in pancreatic islets have been shown to be involved in the expression of *Mafa*, *Ins1*, *Pdx1* and *Glut2* [121]. Calcium-related factors such as CAMK2G is also reported to regulate *Mafa* expression in human islets [122,123]. These results offer clues about how to improve *Mafa* expression and β-cell function in individuals with diabetes.

## 6. Role of MAFA in Human β-Cells

Recent studies also indicate that there are hormone-negative “empty” endocrine cells in human islets with diabetes [124,125], which may be critical for the pathology of β-cell dysfunction. Although it is impossible to perform lineage tracing studies of human β-cells in vivo, researchers have used the viral-induced Cre-loxP system to conduct it. The results of this study demonstrated dedifferentiation followed by transdifferentiation into glucagon-expressing cells of human β-cells in vitro [69,126,127,128]. Because the morphology of murine pancreatic islets is different from that of humans [129], transcription factors that are critical for the function and maintenance of mature β-cells may differ between mice and humans. MAFA is expressed in human β-cells [60,124,125], although MAFA does not peak in humans until 9 years of age [75]. Meanwhile, most results of human studies have shown that MAFB is expressed in not only α-cells but also β-cells in adult human pancreas [60,130,131]. Single-cell RNA-Seq of human islet cells revealed that subpopulations of islet cells coexpressing *MAFA* and *MAFB* represent highly functional and mature subpopulations of β cells [132]. The expression of *MAFA* in β-cells is impaired in humans with diabetes [60,133]. ATAC-seq analysis identified significantly enriched motifs for MAFA in the open chromatin regions of non-diabetic human islets [134]. Recent exome sequencing of a pedigree with autosomal dominant inheritance of diabetes mellitus or insulinomatosis revealed a disease-causing missense mutation of *MAFA*, p.Ser64Phe in two unrelated families. This mutation impairs phosphorylation within the transactivation domain of MAFA and affects its protein stability. In these families, diabetes occurred more often in males, which is consistent with the phenotype of mice carrying the S64F mutation in *Mafa*, which shows accelerated senescence and β-cell dysfunction in males. Meanwhile, insulinomatosis was frequently found in females of these families, which may be related to the fact that the large Maf family of transcription factors was originally identified as oncogenes. [135,136]. Among the factors mentioned thus far, T3 has been shown to stimulate *MAFA* expression in human endocrine precursors, which results in the formation of functional β-cells [113,114,115]. These results suggest that MAFA is important in human β-cells, although other studies have shown that NKX6-1, which is another transcription factor critical for mature β-cells in mice, may be a better indicator of human β-cell function [60,133].

## 7. Conclusions

In summary, accumulating evidence of recent results suggests that MAFA is indispensable for the maturation, function and maintenance of the β-cell in the adult pancreas. Loss of MAFA, which is observed in β-cells of diabetic mice, results in a deeper loss of mature β-cell phenotype and in dedifferentiation with upregulation in the expression of MAFB in β-cells. Inducing upregulation or inhibiting downregulation of MAFA can improve adult β-cell function and can induce the terminal differentiation of β-cells. Thus, MAFA can be a therapeutic target to maintain the function of β-cells. The identification of small molecules to stimulate MAFA expression or function may be useful for the maintenance of normal β-cells or improvement of compromised β-cells and thus would be beneficial for the development of novel strategies for the treatment of diabetes.

Future studies should identify other factors that are critical for the homeostasis of β-cells, which can be demonstrated by showing the conversion of pancreatic endocrine cell fate with adult β-cell-specific genetic engineering. Implication of cell fate conversion in the disease mechanism should be clarified by analysis of diabetic mice and humans. Elucidation of the implication of the β-cell dedifferentiation phenomenon in clinical intensive insulin therapy may result in the development of preemptive medical care for patients with diabetes.

## Figures and Tables

**Table 1 ijms-23-04478-t001:** Genetic lineage tracing studies of transgenic mice to show adult β-cell dedifferentiation.

References	Genes	Mice	Insulin ^(−)^ β-Cells	Upregulated Genes	Trans- Differentiation	β-Cell Death
Talchai et al. Cell 2012	*Foxo1*	*RIPCre;Foxo1^fl/fl^;RosaEGFP* with metabolic stress	Detected	Neurog3, Oct4, Nanog	β- to α, δ, γ--cells	Similar to the controls (TUNEL, cleaved caspase-3)
Gao et al. Cell Metab 2014	*Pdx1*	*RIPCreER;Pdx1^fl/fl^;RosaYFP*	Detected	Mafb, Gcg	β- to α-cells	Not marked (Cleaved caspase-3)
Wang et al. Cell Metab 2014	*Kcnj11* (K_ATP_-GOF *)	*RIPCre or Pdx1CreER;RosaKir6.2* [K185Q,DN30] IRES-GFP	Detected	Neurog3	β- to α-cells	No significant difference (TUNEL, cleaved caspase-3)
Nishimura et al. Diabetologia 2015	*Mafa*	*Mafa^-/-^;RIPCreER;RosaYFP*	Detected	Neurog3, Mafb, Mct1	β- to α-cells	No significant difference (TUNEL)
Ahmad et al. PLoS ONE 2015	*Pax6*	*RIPCreER;Pax6^fl/fl^;RosaYFP*	Detected	Ghrl	β- to ε-cells	Not affected (TUNEL)
Ediger et al. J Clin Invest 2017	*Ldb1*	*MIPCreER;Lbd^fl/fl^;* *RosaYFP*	Detected	Neurog3, Rfx6	(-)	No change in islet size and density
Gutiérrez et al. J Clin Invest 2017	*Nkx2-2*	*RIPCre;Nkx2-2^fl/fl^;RosaTomato*	Detected	Ppy, Sst, Acot7	β- to α, δ, γ-cells	Little evidence (Cleaved caspase-3)
Lee et al. Diabetologia 2022	*Xbp1*	*Pdx1CreER;Xbp1^fl/fl^;RosaGFP* with metabolic stress	Detected	Arx, Irx2, Gcg	β- to α-cells	Significantly increases (TUNEL)

* Gain-of-function mutation.

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
