# Peer review of "Role of the Transcription Factor MAFA in the Maintenance of Pancreatic β-Cells"

_ijms, 2022, doi:10.3390/ijms23094478_

Round 1
Reviewer 1 Report
This is an interesting, timely and detailed review on the role of an important transcription factor, MafA, in insulin turnover and its pathophysiological relevance specifically to T2DM. I have only two, minor suggestions for the authors that may increase the visibility and impact of this interesting article:
- A summary graphic capturing main signaling nodes and downstream targets of MafA and its main functions described in this review would be useful.
- The role of redox stress in regulation of MafA expression (described in section 5.1) lacks full mechanistic details. In particular, the seminal role of the crucial redox protein, TXNIP, in regulation of beta-cell redox state and MafA expression and its implication for T2DM was not discussed. This should be also reviewed.
Author Response
Reviewer: 1
Comments and Suggestions for Authors:
This is an interesting, timely and detailed review on the role of an important transcription factor, MafA, in insulin turnover and its pathophysiological relevance specifically to T2DM. I have only two, minor suggestions for the authors that may increase the visibility and impact of this interesting article:
We appreciate the reviewer for careful and considerate evaluation of our work and for constructive comments to improve our manuscript.
- A summary graphic capturing main signaling nodes and downstream targets of MafA and its main functions described in this review would be useful.
We thank the reviewer and agree that suggested figure would be useful. However, because some of upstream pathways and downstream targets of MafA are still controversial, we would like to mention these in the main text with references to avoid misunderstanding. We changed Table 1 to figure to easily recognize genes downregulated in Mafa-deficient islets.
- The role of redox stress in regulation of MafA expression (described in section 5.1) lacks full mechanistic details. In particular, the seminal role of the crucial redox protein, TXNIP, in regulation of beta-cell redox state and MafA expression and its implication for T2DM was not discussed. This should be also reviewed.
Txnip is initially mentioned in the section “5.2 Noncoding RNA” with miR-204. I agree with the reviewer that Txnip should be mentioned as an important redox regulator for β-cells and it is now described in lines 221-226 of the section 5.1.
Page 6, Lines 221-226:
“Besides c-JUN, which is induced by oxidative stress, another important factors that can deteriorate β-cell function via downregulation of MAFA during diabetes is TXNIP, a cellular redox regulator. TXNIP is upregulated in β-cells during diabetes and has been shown to induce miR-204 by inhibition of STAT3 phosphorylation, which in turn directly inhibits MAFA and insulin expression [86].”

Reviewer 2 Report
The manuscript by Nishimura comprehensively comments on the role of MafA in preserving a functional pancreatic beta-cell by reviewing the most significant publications. This review paper is clear, well structured, and easy to follow. There are some issues that if addressed would substantially strengthen the manuscript. The reviewer encourages the author to consider citing the publications listed below.
- The author may need to comment on different Mafa transgenic animal models, and the induced phenotypes, as well as the influence on key beta-cell genes.
- In Line 58-62, Mafb also binds to insulin (PMID: 17360442) and Mafa (PMID: 17360442, PMID: 20584984), and works together with Pdx1, Neurod1, during beta-cell development.
- In Line 81-85, the author should also add studies in Mafb KO mice.
- In Line 87, please mention the starting time point of the expression of Mafb during mouse embryonic development and the corresponding reference.
- In Line 98-100, more evidence is required, please refer to PMID: 26696512 and PMID: 21190012.
- In Line 86, please double check the initial appearing time of Mafa, E12.5 or E13.5.
- In Line 80, the title maybe not proper since only Mafb, not Mafa, is required during mouse beta-cell development. Pancreas development is not affected in Mafa overall knockout mice and pancreatic-specific Mafa knockout mice. Mafa mainly contributes to beta-cell maturation and function after birth. The effect of Mafa on beta-cell gene transcription during development is only seen in the Mafb knockout mice as compensation.
- On Page 5, the fourth column in Table 2 is not clearly labelled. The title is Insulin (-) beta-cells with the content as (+). The reviewer assumes the author meant that whether the Insulin-negative beta-cells were detected in different transgenic mouse models.
- In Section 5.3, the fact that Mafb binding to Mafa promoter region should be addressed.
- In Section 5.5, the regulation of Calcium-related factors on Mafa expression should be added. For instance: 1). It has been documented that CaMKII activation affects Mafa expression in beta-cells (PMID: 24944908, PMID: 18948074). 2). A study recently reported a negative feedback that Mafa is a downstream target of the Calcium channel subunit gamma4 (doi.org/10.3390/biomedicines10040770).
Author Response
Reviewer: 2
Comments and Suggestions for Authors
The manuscript by Nishimura comprehensively comments on the role of MafA in preserving a functional pancreatic beta-cell by reviewing the most significant publications. This review paper is clear, well structured, and easy to follow. There are some issues that if addressed would substantially strengthen the manuscript. The reviewer encourages the author to consider citing the publications listed below.
We would like to thank the reviewer for the detailed reading of our manuscript and for the excellent and thoughtful suggestions by kindly providing PMID. We have addressed all the comments and suggestions raised by the reviewer, which we believe improved our manuscript.
- The author may need to comment on different Mafa transgenic animal models, and the induced phenotypes, as well as the influence on key beta-cell genes.
Comment on different Mafa transgenic mice, induced phenotypes and their gene expression are now described in lines 77-83 of the section 2.
Page 2, lines 77-83:
So far, the analysis of global [12,29,30] and pancreas-specific Mafa knockout mice [31] has been reported. These mice show similar phenotype, showing impaired mass and function of β-cells by 3 to 4 weeks of age, reduced proliferation with no accelerated apoptosis of β-cells, downregulation in expression of Ins1, Ins2, Slc2a2, Slc30a8 and Pdx1 in their islets, and glucose intolerance. Transcriptome analyses of islets isolated from Mafa knockout mice have further elucidated candidate molecules that are regulated by MAFA. Genes downregulated in both knockout mice are summarized in Figure 1 [30,31].
- In Line 58-62, Mafb also binds to insulin (PMID: 17360442) and Mafa (PMID: 17360442, PMID: 20584984), and works together with Pdx1, Neurod1, during beta-cell development.
Important role of Mafb on the insulin and Mafa promoter, suggested by the reviewer, is now mentioned in lines 98-100 of the section 3 to describe role of Mafb in development of pancreas.
Page 3, lines 98-100:
Interestingly, MAFB shares a DNA binding region with MAFA and thus can bind the C1-A2 element of insulin promoter in vivo, activate it in vitro, and can also bind R3 of Mafa promoter during β-cell development [45,46].
- In Line 81-85, the author should also add studies in Mafb KO mice.
Studies in Mafb KO mice are described in lines 120-122 of the section 3.
Page 3, lines 120-122:
Mafb-deficient pancreas have a reduced number of insulin+ and glucagon+ cells with reduced expression of PDX1 and MAFA, without affecting endocrine progenitor cells expressing NEUROG3, NKX2-2, NKX6-1 and PAX6 [45,50].
- In Line 87, please mention the starting time point of the expression of Mafb during mouse embryonic development and the corresponding reference.
The starting time point of the expression of Mafb during mouse embryonic development is mentioned in lines 95-98 of the section 3. Moreover, the expression pattern of Mafb is described in lines of 100-104 of the section 3.
Page 3, lines 95-98:
Meanwhile in adult mice pancreas, another large Maf factor MAFB is expressed in glucagon+ cells as early as at E10.5 and also in insulin+ cells prior to MAFA during embryonic development of murine pancreas [43,44].
Page 3, lines 100-104:
MAFB expression persisted in glucagon+ and insulin+ cells but not in either somatostatin+ or pancreatic polypeptide+cells during development, which is gradually restricted to glucagon+ cells after birth, and is selectively expressed in the glucagon-producing α-cells of the adult pancreatic islets [43,45,47].
- In Line 98-100, more evidence is required, please refer to PMID: 26696512 and PMID: 21190012.
We deeply appreciate the reviewer to suggest references, which are cited in lines of 111-113 of the section 3.
Page 3, lines 111-113:
These observations are further supported by the results of stem cell studies showing that the ability to secrete insulin from ES-derived insulin+ cells accompanied the expression of MAFA [51-53].
- In Line 86, please double check the initial appearing time of Mafa, E12.5 or E13.5.
The initial appearing time of Mafa is revised in lines 93-95 of section 3.
Page 3, lines 93-95:
The expression of MAFA occurs during pancreatic development starting at E12.5 to E13.5 and can be observed exclusively in insulin-expressing (insulin+) cells [43,44].
- In Line 80, the title maybe not proper since only Mafb, not Mafa, is required during mouse beta-cell development. Pancreas development is not affected in Mafa overall knockout mice and pancreatic-specific Mafa knockout mice. Mafa mainly contributes to beta-cell maturation and function after birth. The effect of Mafa on beta-cell gene transcription during development is only seen in the Mafb knockout mice as compensation.
We agree with the reviewer’s suggestion and changed the title of section 3, which is in line 88.
Page 3, line 88:
- The role of Maf factors in the developing pancreas
- On Page 5, the fourth column in Table 2 is not clearly labelled. The title is Insulin (-) beta-cells with the content as (+). The reviewer assumes the author meant that whether the Insulin-negative beta-cells were detected in different transgenic mouse models.
The reviewer's assumption is correct and the content is revised in the Table, from “(+)” to “Detected”.
- In Section 5.3, the fact that Mafb binding to Mafa promoter region should be addressed.
Now Mafb binding to Mafa promoter region is mentioned in lines 262-264 of section 5.3.
Page 7, lines 262-264:
The Mafa promoter is regulated by region 3, which is located -8118 to -7750 bp from the transcription start site, where PDX1, FOXA2, NKX2-2, PAX4, PAX6, NKX6-1, NEUROD1, ISL1 and MAFB can bind [105].
- In Section 5.5, the regulation of Calcium-related factors on Mafa expression should be added. For instance: 1). It has been documented that CaMKII activation affects Mafa expression in beta-cells (PMID: 24944908, PMID: 18948074). 2). A study recently reported a negative feedback that Mafa is a downstream target of the Calcium channel subunit gamma4 (doi.org/10.3390/biomedicines10040770).
The regulation of Calcium-related factors on Mafa expression is now mentioned in lines 300-301 of section 5.5.
Page 8, lines 300-301:
Calcium-related factors such as CAMK2G is also reported to regulate Mafa expression in human islets [123,124].
